# The dual mechanism of social networks on the relationship between internationalization and firm performance: Empirical evidence from china

Xin Cao[1,2], Peng Li[3]*, Xiaozhi Huang[3]*, Limin Fan[3]

**1** School of Economics, Guangxi University, Nanning, China, **2** School of Economics and Trade, Guangxi University of Finance and Economics, Nanning, China, **3** School of Business Administration, Guangxi University, Nanning, China

* lp35682@163.com (LP); hxiaozhi@mail3.sysu.edu.cn (HX)

**Data Availability Statement:** All relevant data are within the paper and Supporting Information files.

**Funding:** This work was supported by the National Social Science Key Foundation of China (17AJL012),the National Natural Science

## Abstract

The effects of social networks on the relationship between internationalization and firm performance have been well documented in the international literature, and two dimensions of social networks have also been identified: business ties and political ties. However, few efforts have been made to examine whether there are different mechanisms of business ties and political ties. Based on social network theory and boundary spanning theory, we build a model of a dual mechanism of social networks, and the business ties and political ties of social networks that correspond with information processing and the external representation of boundary spanning theory. Using the data of Chinese listed companies in 2005–2013 and 2013–2017 to test the model, the results indicate that (1) in the relationship between internationalization and firm performance, the role of social networks has a dual mechanism. (2) Business ties play a mediating role in the relationship between internationalization and firm performance. Business ties are conducive to a company's information acquisition and knowledge sharing and play the role of information processing. (3) Political ties play a U-shaped moderating role in the relationship between internationalization and firm performance and assume the role of external representation.

## 1 Introduction

In recent years, an increasing number of companies from emerging markets have been integrating into international markets and competing internationally to enhance their competitive advantages [1]. As latecomers to the world economy, emerging market companies seem to develop in the midst of contradictions. On the one hand, compared with multinational companies in developed markets, emerging market companies have a large gap in company size, management knowledge, advanced technology and so on [2–4] and a relative lack of knowledge about their potential international expansion host countries [4]. On the other hand, companies in emerging markets expand rapidly in the international market. They invest in

Foundation of China (72062001,71872055), the Open subject of first-class discipline (cultivation) of Applied Economics in Guangxi(2018ZD15).

**Competing interests:** The authors have declared that no competing interests exist.

overseas areas that are different from the institutional environment of their home countries and alleviate their disadvantages as latecomers by searching for international knowledge and institutional support [3, 5]. This contrast not only introduces challenges to the study of emerging market multinationals but also inspires scholars' interest in the study of this area. The challenge is that companies expect to improve performance through international expansion, but in the process of internationalization companies may encounter difficulties in addition to the pursuit of potential gains [6]. An important reason for these difficulties is that companies lack corresponding international experience and knowledge [3, 4], so it seems necessary to study the mechanisms of emerging market multinational companies to improve performance. Interestingly, how emerging market multinationals accumulate international knowledge through overseas expansion to mitigate their latecomer disadvantages and improve firm performance has become a subject of debate for multinational enterprises [7–9].

In previous studies, scholars have found both a linear and nonlinear relationship between internationalization and the performance of multinational enterprises [3, 10, 11]. These findings have prompted scholars to research the question of how internationalization leads to superior performance of multinational enterprises. Among them, the literature based on the perspective of social networks argues that social networks can provide companies with key resources and information, enhance trust between companies [12–15], make up for the lack of international knowledge and experience, and improve their performance [16–18]. Some studies have analyzed the impact of social networks on firm performance at the single dimension level [18], however, more studies have discussed the impact of social networks on firm performance from the two dimensions of business ties and political ties along the technical route of Peng and Luo (2000). These studies have found that the micro ties of business and politics jointly promote the macro performance of the organization [19], improve the innovation of the enterprise [20], and promoted the overseas expansion of the enterprise [21]. The common feature of these studies is that business ties and political ties have the same positive impact on organizational performance. On this basis, scholars have further raised the issue of the difference in the role of business ties and political ties on firm performance [22]. Studies have found that both business ties and political ties play a moderating role in the relationship between enterprise strategy and internationalization but that the intensity of the role is different [23]. Both business ties and political ties can promote enterprise innovation, but they are antagonistic to each other [24]. There are also differences between business ties and political ties in the process of enterprises resolving overseas disputes [25]. This evidence shows that there is a dual influence of business ties and political ties on enterprises [24]. However, there is less research that attempts to answer the question of why there is a dual impact of business ties and political ties on enterprise performance. In view of this research gap, this paper discusses the source of the difference between business ties and political ties from the perspective of influence mechanisms and proposes the following research questions: In the relationship between internationalization and performance, do social networks have a dual mechanism? Why is there a dual mechanism? These questions articulate the view that there are differences between business ties and political ties and motivate the existing studies that focus on the differences in the intensity of social networks' roles to discuss the differences in their influence mechanisms. This particular discussion can provide new explanations for the relationship between internationalization and firm performance from the perspective of the differences in the impact mechanisms of different dimensions.

This paper is based on social network theory [13, 26] and boundary spanning theory [27] to answer the above questions. According to social network theory, this paper divides social networks into business ties and political ties. According to boundary spanning theory, the boundary spanner has two main boundary spanning roles. One is information processing, which

refs to the exchange and sharing of information between the boundary spanner and those outside the company. The other is external representation, which refers to the role of the boundary spanner as a lubricant and is conducive to a company's response to changes in the external environment [27, 28]. Social network theory believes that economic behavior is embedded in interpersonal ties and relationships, and the social network of corporate managers can be divided into business ties and political ties. In view of boundary spanning theory, corporate managers are boundary spanners [29]. Therefore, in the interpersonal network of corporate managers, business ties and political ties can assume the role of boundary spanning. We synthesize social network theory and boundary spanning theory and argue that there is a dual impact mechanism of social networks in the relationship between internationalization and performance. Business ties mainly assume the role of information processing and are the mediating mechanism in the relationship between internationalization and performance, while political ties mainly assume the role of external representation and are the moderating mechanism in the relationship between internationalization and performance. To refine the impact of internationalization on performance, this paper follows the existing research trends and distinguishes internationalization into external internationalization orientation and internal internationalization orientation [18, 30] and examines the issue of the dual mechanism of social networks in the relationship between the two internationalization orientations and performance.

This paper tests our hypothesis of a dual mechanism of social networks based on data from Chinese listed companies, and the results of the study show that outward internationalization orientation positively affects firm performance through business ties, while political ties play a U-shaped moderating role in the relationship between internationalization and performance. The main contributions of this paper are as follows: (1) The proposal of the dual mechanism of social networks has made an incremental contribution to the social networks literature. We find that in the relationship between internationalization and firm performance, business ties play the role of a mediating mechanism, while political ties play the role of a moderating mechanism. The findings take the existing research on the differential role of social networks a step further in terms of differences in the intensity [22, 23] and the differences in the mechanism; thus, the current paper makes an incremental contribution to the research of social networks. (2) Our research enriches the boundary spanning theory. We connect the business ties and political ties in social networks with information processing and external representation in boundary spanning theory and use boundary spanning theory to construct a theoretical model of the dual mechanism of social networks. Based on boundary spanning theory, this paper responds to the discussion of the dual impact of literature on social networks [24], and contributes to boundary spanning theory. (3) Our study provides new evidence and research perspectives on the relationship between multinational enterprises' internationalization and firm performance in emerging economies. We empirically test the theoretical model based on Chinese listed companies, and the empirical results support the hypothesis that social networks have dual mechanisms. At the same time, the role of informal institutions has always been a subject of debate for scholars in the study of transnational corporations in emerging economies. The dual mechanism model of social networks proposed by our research provides a new perspective for the study of transnational corporations in emerging economies.

## 2 Literature review

### 2.1 Internationalization orientation

The international expansion of a company can usually be guided by two internationalization strategies: outward internationalization and inward internationalization [18, 30, 31]. The

outward internationalization orientation refers to local firms engaging in transactions in overseas markets [30], seeking business opportunities in overseas markets, developing overseas business, and making direct investments overseas [18]. An inward internationalization orientation refers to local firms engaging in transactions with overseas firms in their home countries [30], taking advantage of new technologies and management skills in overseas markets, and receiving foreign direct investment from overseas companies in local firms [18].

An outward internationalization orientation strategy is conducive to finding potential market opportunities, understanding the new needs of the international market, and promoting the expansion of the geographical scope of internationalization to enhance the company's scale and scope [32]. An inward-oriented internationalization strategy is mainly to improve company performance by gaining advanced management experience and obtaining technology from abroad [33]. Local companies will consume and absorb the knowledge and experience they learn from abroad to develop their own capabilities, thereby strengthening their competitive advantage and improving their performance [18].

## 2.2 Perspective of boundary spanning

To survive and maintain growth, organizations need to rely on the external environment to find critical resources and business opportunities. According to boundary spanning theory [27], organizations rely on boundary spanners to ensure that social and economic exchanges between organizations and between organizations and their external environment can be facilitated smoothly, thus ensuring that organizations are protected from disruptive external environmental factors. To achieve this goal, boundary spanners need to take on two important roles, namely, information processing and external representation [27].

When assuming the role of information processing, the boundary spanner obtains information from the external environment of the organization and then provides information to users within the organization through coding, filtering and translation processes [27]. The boundary spanner will also share appropriate internal information with other organizations. Through the two paths of external information inflow and internal information sharing, the boundary spanner becomes a bridge connecting the organization and the external environment [34]. In the exchange relationship between organizations, communication is a process of information transmission, which has become the link between the two sides of the exchange [35].

Boundary spanners perform the role of external representation by promoting resource sharing, expressing perceptions and expectations of the external environment, providing coordination and assistance to the external environment, and achieving common goals through linkages between boundary spanners and organizations [27]. Through boundary spanners, organizations are connected with each other to form synergy to achieve common goals. In the exchange relationship between organizations, cooperation refers to the fact that both sides of the exchange achieve the final result through similar or complementary collaborative actions [36]. On the one hand, both sides of the exchange are aware that cooperation has become increasingly important with the enhancement of mutual relations between the two sides; on the other hand, when an unexpected situation occurs or there is a conflict between the two sides of the exchange, the boundary spanners are needed as a channel by which to solve the problem through reasonable negotiation, persuasion and joint action [37].

## 2.3 Social networks and corporate internationalization

One of the basic viewpoints of social network theory is that economic behavior is embedded in the interpersonal network. Based on this concept, many studies emphasize the importance of

the interpersonal network established by corporate managers and external personnel for enterprises [22]. The literature identifies two types of important interpersonal networks of corporate managers: political ties and business ties. Political ties refer to the relationship between corporate managers and government staff at different levels, and business ties refer to the relationship between corporate managers and their business partners [19, 22]. The common feature of these two ties is that they rely on social networks rather than formal institutions to obtain resources, but the types of resources obtained by the two ties are different [22]. Research has found that business ties allow resource and information sharing, coordinate logistics, reduce distribution costs and limit the opportunistic behavior of partners [35]. The benefits of political ties include valuable market information, fewer bureaucratic delays and monetary and nonmonetary incentives [38].

To explain the complex relationship between corporate internationalization and firm performance, scholars have introduced social network theory into the study of corporate internationalization, arguing that companies can make up for their lack of international knowledge and experience in the internationalization process by embedding themselves in social networks, which can help companies identify opportunities in the international market and enhance their connections with overseas institutions through social network connections. The positive impact of social networks on corporate internationalization performance has also been supported by empirical research [39]. Of course, the literature also suggests that there may be a negative aspect of social networks in the internationalization of firms. One study investigates the impact of local market political ties and local formal institutions on the internationalization of emerging market firms in the Chinese context based on resource dependency theory and an institutional base view. It finds that local political connections may prevent firms from emerging markets from implementing international strategies by reducing dependency constraints imposed by local governments and foreign firms and that the negative impact can be mitigated by the establishment of formal institutions [40].

From the above literature review, it can be seen that the existing research on the relationship between corporate internationalization, social networks and firm performance has achieved abundant results, but there are still deficiencies in the research perspectives. On the one hand, the existing research on outward and inward internationalization orientation mainly focuses on the relationship between the two internationalization strategies [30, 41, 42], and the focus of discussion is the influence of inward internationalization strategy on outward internationalization strategy, and less frequently discusses the impact of both internationalization strategies on firm performance based on a social network perspective. On the other hand, the existing literature on the role of social networks in the process of corporate internationalization distinguishes the role of business ties and political ties in terms of size, but it is based less on the mechanism of action to analyze the difference between business ties and political ties in corporate internationalization [22]. Based on this understanding, we propose the dual mechanism of social networks in the relationship between internationalization and firm performance and divide the internationalization strategy into outward internationalization orientation and inward internationalization orientation. We believe that in the relationship between internationalization and firm performance, business ties play a mediating role, while political ties play a moderating role.

## 3 Research hypothesis and conceptual model

### 3.1 Research hypothesis

**3.1.1 The mediating role of business ties.**    For companies that choose outward internationalization orientation, there is a strong driving force for embedding in social networks

because the business ties in social networks can provide convenient connections for companies to connect with foreign markets and play a role as a bridge link [43], providing key resources for the internationalization of companies, such as obtaining information about foreign market opportunities, gaining foreign experience and enhancing trust. These mechanisms are essential for outward international companies to mitigate information and knowledge barriers and improve the quality of their export business. Business ties can enhance the trust relationship between companies and their foreign partners. This trust mechanism based on relational networks can reduce international transaction costs and help companies succeed in the international market [44]. In general, the role of business ties is reflected in the following three aspects. First, business ties give companies product information, information about market events and changes, and information about the reliability of the people they deal with, which are difficult to obtain in the formal market system [22]. Second, strong social connections between a company and other business partners can enhance the company's ability to learn and facilitate the flow of knowledge and technology between networks [45]. Finally, business ties promote trust between firms and legitimize their presence in business communities [46]. These important market resources can increase the attraction of market transactions between companies and business partners, promote trade between companies, and bring economic benefits to companies. Therefore, the following hypothesis is proposed:

H1: Outward internationalization orientation positively affects firm performance through business ties.

The impact mechanism of business ties on the relationship between inward internationalization orientation and performance is similar to that of outward internationalization. The complex domestic market environment has brought development constraints to inward international orientation in emerging economies. Business ties are an important way for companies to obtain information and knowledge to develop organizational capabilities [47]. Therefore, business ties embedded in informal institutions can increase the probability of success for inward international-oriented companies [18].

Based on this view, we believe that just as business ties can bring information and resources to companies with an outward internationalization orientation, business ties can also bring information and resources to companies with an inward internationalization orientation: foreign products or technical information suitable for the domestic market; learning from advanced foreign management experience to improve the ability to identify market opportunities; and building trustful relationships to access the technology, capital and management experience of foreign partners. Based on this, the following hypothesis is proposed:

H2: Inward internationalization orientation positively affects firm performance through business ties.

**3.1.2 The moderating role of political ties.**   Political ties refer to the informal social connection between the company and the local government departments and management institutions. Similarly, political ties can bring critical regulatory resources to companies. First, governments in transition economies usually guide economic behavior through industrial development planning and economic policies. Informal social ties between companies and governments can help companies learn about important policy information. Second, when government agencies control important resources, such as land, bank credit and tax policies, political ties with governments can provide these scarce resources for companies. Finally, political ties enhance the political legitimacy of a company. These key regulatory resources enhance firm performance [22].

The role of political ties is different from that of business ties. The main purpose of business ties is to maximize profits, and companies in business ties share common goals, which can encourage long-term cooperation among companies. In contrast, political ties lack an effective

mechanism to maintain long-term cooperation [22]. In the political ties between a company and a government, government officials are more concerned about the promotion of their official career, while corporate managers are more concerned about the economic returns of the company. This divergence makes it difficult for the company and the government to establish long-term informal social relations. In addition, government officials often change positions and serve in different departments and different places, which also limits the willingness of corporate managers to maintain long-term social ties with government officials [22].

By incorporating boundary spanning theory into the analysis, we argue that the political legitimacy provided by political ties for a company can serve as an external representation of the company's boundary spanner, indicating that the company has a scarce intangible asset and therefore can promote the internationalization expansion of the company [48]. In this case, political ties enhance the positive impact of corporate internationalization on performance. However, at the same time, the literature also noted the negative effects of political ties. For example, the literature has found that political ties will prevent the international expansion of companies [40]. In this case, political ties weaken the positive impact of company internationalization on performance. This shows that the role of political ties is more complex and not simply linear but nonlinear. According to the literature, the negative effects of political ties are mainly reflected in the differences between government officials and corporate management objectives and the lack of effective mechanisms for long-term cooperation [22]. Therefore, we infer that in the early stage of a company's political relationship, it is difficult for government officials and corporate managers to reach common goals, and the benefits brought by the fostering of political relations cannot offset the costs. In this situation, political ties will weaken the positive effect of corporate internationalization on performance. With the continuous enhancement of a company's political ties, as a company's boundary spanner, the company's managers will seek to achieve synergies with external organizations to reach common goals [49]. At this time, political ties can enhance the positive impact of corporate internationalization on performance. Based on the above analysis, political ties will play a U-shaped moderating role in the relationship between corporate internationalization and performance. Therefore, the following hypotheses are proposed:

H3: Political ties play a U-shaped moderating role in the relationship between outward international orientation and performance. Specifically, when political ties develop from a weak level to a medium level, the positive impact of outward international orientation on performance is weakened. When political ties develop from a middle level to a high level, the positive impact of outward internationalization orientation on performance is enhanced.

H4: Political ties play a U-shaped moderating role in the relationship between inward international orientation and performance. Specifically, when political ties develop from a weak level to a medium level, the positive impact of inward international orientation on performance is weakened. When political ties develop from a middle level to a high level, the positive impact of inward internationalization orientation on performance is enhanced.

## 3.2 Conceptual model

We explain the relationship between internationalization and performance based on social network theory and boundary spanning theory. According to boundary spanning theory, corporate managers are important boundary spanners [29], so when corporate managers are building individual social networks, they actually perform the boundary spanning role described in boundary spanning theory [50]. Thus, the actions of corporate managers in establishing business ties and political ties correspond to the two roles of information processing and external representation performed by boundary spanners [50].

When discussing business ties, most of the existing literature argues that business ties facilitate the information acquisition and knowledge sharing of firms [22], which is consistent with the role of information processing in boundary spanning theory, while political ties provide firms with political legitimacy [22] and the protection of companies in the face of external threats. If companies have good government relations in the process of internationalization, this is conducive to promoting their internationalization strategy [48], which conforms to the role of external representation in boundary spanning theory. A company's internationalization strategy can increase its internationalization knowledge and experience through the information processing process of information acquisition and knowledge sharing to promote the improvement of performance. Therefore, business ties play a bridging role similar to the connection between internationalization and performance.

Political ties represent more of a scarce intangible asset for companies; for example, company managers accompanying their national officials on visits can advance the companies' international expansion [48]. In the relationship between internationalization and performance, the role of political ties is more of an enhancement mechanism, and there is a difference between the role of political ties and business ties, not only in the size and direction of the role but also in the mechanism of the role. A firm's internationalization strategy affects performance through business ties, which act as a mediating mechanism; political ties may enhance the relationship between firm internationalization and performance, which act as a moderating mechanism. We call this differentiated role of business ties and political ties the dual mechanism of social networks. Based on this, we construct the conceptual model of this paper.

According to the existing literature, corporate internationalization is divided into outward internationalization orientation and inward internationalization orientation [18, 30]. This paper analyzes the impact of two types of internationalization orientation on firm performance. In the model, the dual mechanism of social networks is introduced, suggesting that business ties mediate the relationship between internationalization orientation and firm performance, while political ties moderate the relationship between internationalization orientation and firm performance. The conceptual model is shown in Fig 1.

## 4 Research method

### 4.1 Econometric model

This paper mainly studies the dual mechanism of social networks in the relationship between internationalization and firm performance, in which business ties are the mediating

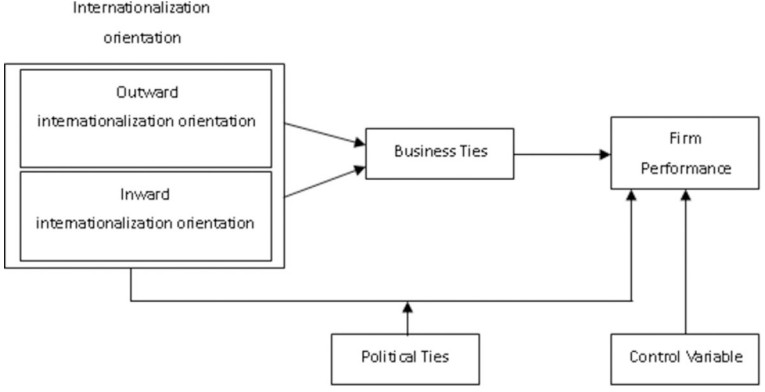

**Fig 1. Conceptual model diagram.**

mechanism and political ties are the moderating mechanism. Therefore, an econometric model to test the mediating effect and moderating effect is constructed.

$$Y_i = \beta_0 + \beta_1 \text{Foreign\_income\_log}_i + \beta_2 \text{FOREIGN}_i + \beta_3 \text{bus\_ratio}_i + \gamma X_i + \varepsilon_i \qquad (1)$$

Formula (1) is an econometric model that tests the mediating effect of business ties, where $Y_i$ is the dependent variable firm performance, $\text{Foreign\_income\_log}_i$ is the independent variable outward internationalization orientation, $\text{FOREIGN}_i$ is the independent variable inward internationalization orientation, $\text{bus\_ratio}_i$ is the mediating variable business ties, $X_i$ is a set of control variables at the company level, and $\varepsilon_i$ is the random disturbance term.

$$Y_i = \beta_0 + \beta_1 \text{Foreign\_income\_log}_i + \beta_2 \text{FOREIGN}_i + \beta_3 \text{pc\_ratio}_i + \beta_4 \text{outw\_pc\_log}_i + \beta_5 \text{inw\_pc}_i + \beta_6 \text{outw\_pc2\_log}_i + \beta_7 \text{inw\_pc2}_i + \gamma X_i + \varepsilon_i \qquad (2)$$

Formula (2) is an econometric model that tests the moderating effect of political ties, where $Y_i$ is the dependent variable firm performance, $\text{Foreign\_income\_log}_i$ is the independent variable outward internationalization orientation, $\text{FOREIGN}_i$ is the independent variable inward internationalization orientation, $\text{pc\_ratio}_i$ is the moderating variable political ties, outw_pc_log$_i$ is the first-order interaction between outward internationalization orientation and political ties, inw_pc$_i$ is the first-order interaction between inward internationalization orientation and political ties, outw_pc2_log$_i$ is the second-order interaction between outward internationalization orientation and political ties, inw_pc2$_i$ is the second-order interaction between inward internationalization orientation and political ties, $X_i$ is a set of control variables at the company level, and $\varepsilon_i$ is the random disturbance term.

## 4.2 Analytical method

This paper uses OLS to estimate Formula (1) and Formula (2). Formula (1) mainly tests the mediating mechanism of business ties. To estimate the mediating effect of business ties, the three-step method [51] and bootstrapping method [52] are used at the same time. Baron and Kenny [51] tested the mediating effect mainly through three regressions as follows:

First, the change of independent variable X can significantly explain the change of dependent variable Y; that is, C in Fig 2A should be significantly not equal to zero.

Second, the change of independent variable X can significantly explain the change of mediating variable M; that is, an in Fig 2B should be significantly not equal to zero.

Third, when the mediating variable M is controlled, the influence (C') of the independent variable X on the dependent variable Y should be equal to zero or significantly reduced (C' < < C). At the same time, b should be significantly not equal to zero, as shown in Fig 2C. This result shows that the effect of X on Y is made through M.

The bootstrapping method considers that the existence of a mediating effect is indicated if a×b≠0 in Fig 2C. Therefore, the core of the bootstrapping method is used to test the following

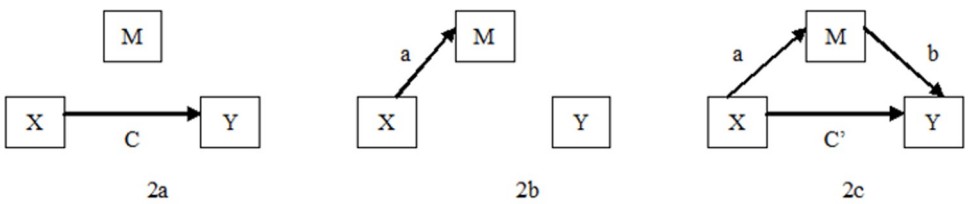

**Fig 2. Three step analysis diagram.**

null hypothesis:

$$H0: \ a \times b = 0 \tag{3}$$

Because in general, we do not know the distribution function of a×b, the null hypothesis cannot be tested directly from an observation sample. The bootstrapping method proposes that multiple repeated self-sampling can be carried out from the original sample to obtain the distribution of a×b; thus, the null hypothesis of Formula (3) can be tested. If the results reject the null hypothesis, then a×b≠0, and a mediating effect exists [52].

Formula (2) mainly tests the moderating effect of political ties. Specifically, this paper hypothesizes that political ties play a U-shaped moderating role in the relationship between outward internationalization orientation and firm performance, as well as the relationship between inward internationalization orientation and firm performance. Therefore, the interaction term between the independent variable and the square term of the moderating variable is constructed to test the U-shaped moderating effect. In Formula (2), $\beta_6$ is the coefficient of the interaction term between outward internationalization orientation and political ties square term, and $\beta_7$ is the coefficient of the interaction term between inward internationalization orientation and political ties square term. If $\beta_6$ is significantly positive, it means that political ties play a U-shaped moderating role in the relationship between outward internationalization orientation and firm performance. If $\beta_7$ is significantly positive, it means that political ties play a U-shaped moderating role in the relationship between inward internationalization orientation and firm performance.

### 4.3 Sample and data sources

We take companies listed on the Shanghai and Shenzhen Stock Exchanges of China as the research samples and analyze them based on data from 2005 to 2013 and 2013 to 2017. The data from 2005 to 2013 are used for hypothesis testing and are obtained from the"China Stock Market & Accounting Research Database"(CSMAR)and Wind databases, where the information on executives of listed companies and foreign shareholdings are obtained from the CSMAR, and the data on foreign shareholdings are manually compiled company by company; the information on overseas sales revenue and financial information are obtained from the Wind database. The final analysis data come from the integration of the above data. First, the information database of listed company executives is constructed, which provides information about listed company executives serving in government agencies or as National People's Congress(NPC)deputies and Chinese People's Political Consultative Conference(CPPCC)members and information about listed company executives serving in other companies. Second, an internationalization database of listed companies is constructed, which collects data on the foreign shareholding of listed companies. Third, this paper constructs the overseas sales income and financial database of listed companies, which provides the corresponding financial analysis data of listed companies. Then, Stata software is used to merge the above three databases and remove the companies in the financial industry, and finally, the analysis data of hypothesis testing in this paper are organized. The data from 2013 to 2017 are in the CSMAR overseas direct investment database, which collects the relevant data of Chinese listed companies in overseas direct investment companies, including executive information, overseas revenue, foreign ownership and financial data, but the starting point of these data is 2013. Therefore, we use these data as additional data and merge them into the data for hypothesis testing from 2005 to 2013 for robustness testing.

### 4.4 Measurement

The dependent variable is firm performance, and ROE is selected as a measure of performance in this paper. The independent variable is internationalization orientation, which is divided

into outward internationalization orientation and inward internationalization orientation. The outward internationalization orientation is measured by the natural logarithm of overseas sales revenue (Foreign_income_log$_i$), while the inward internationalization orientation is measured by the proportion of overseas shareholdings, including overseas institutional shareholdings and overseas individual shareholdings (FOREIGN$_i$).

For social networks, it is necessary to measure business ties (bus_ratio$_i$) and political ties (pc_ratio$_i$). For both types of ties, the measurement method of proportion is adopted; that is, the proportion of people with business relations and political relations in the executive team is used as the measurement. Specifically, for business ties, this paper uses the "Total number of listed companies with concurrent directorships" in CSMAR of personal background characteristics of executives as a statistical basis, which counts the executives of listed companies serving as directors in all other companies (not limited to shareholders). When the indicator is not zero, it means that the executive has a business relationship with other companies. We collated the data and calculated the proportion of executives with business relationships for each listed company, using this proportion to measure the business ties of listed companies. For political ties, we use the "category of government agencies" index in CSMAR of personal background characteristics of executives as a statistical basis, which counts the executives of listed companies as government agents or NPC representatives and CPPCC members. We collated the data and calculated the proportion of each listed company's executive team that has political relationships, using this proportion to measure political ties.

We control for variables such as company size, cash flow, financial leverage, company growth rate and company age. Company size is measured by total assets, cash flow is measured by the ratio of net cash flow generated by business activities to operating income, financial leverage is the liability to asset ratio and the company growth rate is calculated by the annual growth rate of total operating income.

## 5 Results

Two methods are used to analyze the mediating role of business ties. First, the traditional three-step method [51] was applied, the analysis software was Stata12, and the OLS regression method was applied. During the analysis, dummy variables for year and industry were generated to control for the role of year and industry. To control for the influence of heteroscedasticity and sequence correlation, the vce(r) option was used to generate the standard error of correcting heteroscedasticity and sequence correlation. To further verify the mediating role of business ties, we refer to the method of Preacher and Hayes [52] and use bootstrapping to test the mediating role. The analysis software uses SPSS22 and loads Hayes' PROCESS program. For the moderating effect of political ties, hierarchical regression is performed by constructing the interaction term between internationalization and the square of political ties. The analysis software is Stata12, which also controls for the role of industry and year and controls for the effects of heteroscedasticity and sequence correlation. Before reporting the results of the regression analysis, correlation analyses were performed on the main variables, and the results are presented in Table 1.

### 5.1 The results of the mediating effect of business ties

First, OLS is used to analyze the mediating effect of business ties (bus_ratio), and the results are presented in Table 2, where the results for the control industries and years are not reported due to space constraints. The dependent variable is firm performance ROE. Model (1) is the analysis result of the control variables. Model (2) is the result of adding the independent variable of internationalization orientation to Model (1), namely, outward internationalization

**Table 1. Table of correlation coefficients of main variables.**

|  | ROE | Foreig~g | FOREIGN | bus ra~o | pc ratio | TAssets | cash i~e | Aliabi~s | Age | growth |
|---|---|---|---|---|---|---|---|---|---|---|
| ROE | 1 |  |  |  |  |  |  |  |  |  |
| Foreign in~g | 0.0283* | 1 |  |  |  |  |  |  |  |  |
| FOREIGN | 0.0464* | 0.1198* | 1 |  |  |  |  |  |  |  |
| bus ratio | 0.0692* | 0.0633* | 0.0493* | 1 |  |  |  |  |  |  |
| pc ratio | 0.0347* | -0.0243 | -0.0209* | 0.1112* | 1 |  |  |  |  |  |
| TAssets | 0.0775* | 0.4115* | 0.1167* | 0.0832* | 0.0528* | 1 |  |  |  |  |
| cash income | 0.1545* | 0.0222 | 0.0307* | 0.0210* | 0.0506* | 0.0142 | 1 |  |  |  |
| Aliabilities | -0.2016* | 0.2466* | -0.0840* | -0.0502* | -0.00190 | 0.2328* | -0.1142* | 1 |  |  |
| Age | -0.0464* | 0.0816* | 0.0114 | -0.0113 | 0.0193* | 0.0620* | -0.0344* | 0.2736* | 1 |  |
| growth | 0.2631* | 0.00580 | -0.0194* | -0.00480 | -0.00640 | 0.0362* | 0.00850 | 0.0283* | 0.00490 | 1 |

orientation (Foreign_income_log) and inward internationalization orientation(FOREIGN), where outward internationalization orientation has a positive effect on firm performance with a significant result ($\beta = 0.514$, t = 5.18, p<0.001) and inward internationalization orientation has a nonsignificant effect on firm performance ($\beta = 0.249$, t = 0.20, p>0.05). Model (3) takes business ties as the dependent variable, in which outward internationalization orientation has

**Table 2. The mediating effect of business ties.**

|  | (1) | (2) | (3) | (4) |
|---|---|---|---|---|
|  | ROE | ROE | bus_ratio | ROE |
| TAssets | 1.72e-10*** | 1.05e-10*** | 2.99e-13** | 1.02e-10*** |
|  | (13.44) | (6.14) | (3.16) | (6.03) |
| cash_income | 0.0949*** | 0.156*** | 0.000156* | 0.154*** |
|  | (10.94) | (9.14) | (2.14) | (9.02) |
| Aliabilities | -0.151*** | -0.147*** | -0.000199*** | -0.145*** |
|  | (-13.51) | (-9.34) | (-4.21) | (-9.30) |
| Age | 0.0970** | 0.0489 | 0.000594* | 0.0423 |
|  | (3.22) | (1.18) | (2.20) | (1.02) |
| growth | 6.434*** | 7.745*** | 0.00360 | 7.705*** |
|  | (19.60) | (14.56) | (1.64) | (14.48) |
| Foreign_income_log |  | 0.514*** | 0.00186** | 0.493*** |
|  |  | (5.18) | (3.17) | (4.98) |
| FOREIGN |  | 0.249 | 0.0115 | 0.147 |
|  |  | (0.20) | (1.09) | (0.12) |
| bus_ratio |  |  |  | 11.15*** |
|  |  |  |  | (5.17) |
| _cons | -0.723 | 1.563 | -0.0630* | 2.129 |
|  | (-0.15) | (0.55) | (-2.54) | (0.74) |
| N | 9424 | 4927 | 4976 | 4927 |
| r2 | 0.170 | 0.187 | 0.0463 | 0.191 |
| r2_a | 0.167 | 0.183 | 0.0409 | 0.186 |

Note: t statistics in parentheses

* p < 0.05,

** p < 0.01,

*** p < 0.001

a significant positive effect on the mediating variable business ties ($\beta$ = 0.00186,t = 3.17,p<0.01) and inward internationalization orientation has no significant effect on business ties ($\beta$ = 0.0115, t = 1.09,p>0.05). Model (4) puts the independent variable internationalization orientation and the mediating variable business ties into the regression equation at the same time, and the results show that the mediating variable business ties has a significant effect ($\beta$ = 11.15,t = 5.17,p<0.001, and outward internationalization orientation still has a significant effect ($\beta$ = 0.493,t = 4.98, p<0.001, but its effect is smaller than the results of model (2), which indicates that outward internationalization orientation affects firm performance positively through business ties, and H1 is supported. However, the path of inward internationalization orientation affecting firm performance through business ties is not significant, and H2 is not supported.

To further verify the mediating effect, bootstrapping analysis was carried out. The PROCESS program of Hayes was used to select model 4, the confidence interval was set to 95%, and the self-sampling number was 5000. The results found that the mediating role of business ties was significant in the relationship between outward international orientation and firm performance (LLCI = 0.0133, ULCI = 0.0426) with a mediating effect size of 0.0261; after controlling for the role of business ties, the direct role of outward international orientation was not significant (LLCI = -0.0047, ULCI = 0.3171, so H1 is supported. The mediating role of business ties in the relationship between inward international orientation and firm performance is also significant (LLCI = 0.2660, ULCI = 0.7518), with a mediating effect size of 0.4810; after controlling for the role of business ties, the direct role of inward international orientation remains significant (LLCI = 3.8142, ULCI = 9.1378). Business ties play a partial mediating role in the relationship between inward internationalization orientation and firm performance, and the bootstrapping analysis results support H2.

## 5.2 The result of the moderating effect of political ties

OLS is also used to analyze the moderating effect of political ties. The first-order product term (outw_pc_log) of outward internationalization orientation and political ties and the first-order product term (inw_pc) of inward internationalization orientation and political ties are constructed. The second-order product term (outw_pc2_log) of outward internationalization orientation and political ties and the second-order product term (inw_pc2) of inward internationalization orientation and political ties are constructed. Model (1) is the result of controlling variables. Model (2) is the result of model (1) with the addition of independent variables outward internationalization orientation and inward internationalization orientation. Model (3) is the result of model (2) with the addition of political ties and the first-order product term. Model (4) is the result of adding the second-order product term to model (3). Since we need to test the U-shaped moderating effect of political ties, we mainly focus on the result of the second-order product term in model (4), where the result of the second-order product term of outward internationalization orientation and political ties is positive and significant ($\beta$ = 7.433,t = 2.00,p<0.05) and the result of the second-order product term of inward internationalization orientation is marginally significant at the level of 0.1 ($\beta$ = 239.4,t = 1.76, p<0.078), indicating that political ties play a U-shaped moderating role in the relationship between internationalization and firm performance, which is supported by H3 and H4. The results are presented in Table 3.

## 5.3 Robustness analysis

The robustness of the results is tested from three aspects: one is to generate the lag term of the corresponding variables, the second is to analyze the role of social networks in the eastern and western differences, and the third is additional data.

**Table 3. The moderating effect of political ties.**

|  | (1) | (2) | (3) | (4) |
|---|---|---|---|---|
|  | ROE | ROE | ROE | ROE |
| TAssets | 1.72e-10*** | 1.05e-10*** | 1.06e-10*** | 1.04e-10*** |
|  | (13.44) | (6.14) | (6.16) | (6.16) |
| cash_income | 0.0949*** | 0.156*** | 0.157*** | 0.154*** |
|  | (10.94) | (9.14) | (9.16) | (9.15) |
| Aliabilities | -0.151*** | -0.147*** | -0.147*** | -0.145*** |
|  | (-13.51) | (-9.34) | (-9.36) | (-9.32) |
| Age | 0.0970** | 0.0489 | 0.0507 | 0.0485 |
|  | (3.22) | (1.18) | (1.22) | (1.17) |
| growth | 6.434*** | 7.745*** | 7.746*** | 7.697*** |
|  | (19.60) | (14.56) | (14.57) | (14.57) |
| Foreign_income_log |  | 0.514*** | 0.386 | 0.738*** |
|  |  | (5.18) | (1.95) | (3.58) |
| FOREIGN |  | 0.249 | 3.315 | 3.992 |
|  |  | (0.20) | (1.45) | (1.56) |
| pc_ratio |  |  | -13.78 | 50.34 |
|  |  |  | (-0.68) | (1.54) |
| inw_pc |  |  | -20.65 | -66.66* |
|  |  |  | (-1.64) | (-1.97) |
| outw_pc_log |  |  | 0.770 | -3.043 |
|  |  |  | (0.72) | (-1.73) |
| pc2 |  |  |  | -116.6 |
|  |  |  |  | (-1.77) |
| inw_pc2 |  |  |  | 239.4 |
|  |  |  |  | (1.76) |
| outw_pc2_log |  |  |  | 7.433* |
|  |  |  |  | (2.00) |
| _cons | -0.723 | 1.563 | 3.881 | -2.187 |
|  | (-0.15) | (0.55) | (0.89) | (-0.49) |
| N | 9424 | 4927 | 4927 | 4927 |
| r2 | 0.170 | 0.187 | 0.188 | 0.191 |
| r2_a | 0.167 | 0.183 | 0.183 | 0.186 |

Note: t statistics in parentheses

* p < 0.05,

** p < 0.01,

*** p < 0.001

(1) Lag term. The independent variables of outward internationalization orientation and inward internationalization orientation, as well as the lag term of the corresponding control variables in one period, are generated, and the same treatment method is used for OLS analysis. The results show that the mediating role of business ties is still significant in the relationship between outward internationalization orientation and firm performance, with the regression coefficient of outward internationalization orientation decreasing from .239($\beta$ = .239,t = 2.32,p<0.02) before the addition of business ties to .213 ($\beta$ = .213,t = 2.06,p<0.04) after the addition of business ties. In the relationship between outward internationalization orientation and firm performance, the U-shaped moderating effect of political ties is

marginally significant ($\beta$ = 8.33355,t = 1.63,p = 0.103); in the relationship between inward internationalization orientation and firm performance, the U-shaped moderating effect of political ties is significant ($\beta$ = 292.6611,t = 2.43,p<0.05).

(2) Regional differences in the role of social networks between the eastern and western regions. We also analyze the relationship between internationalization orientation and firm performance and the difference in the role of social networks between listed companies in the eastern region and those in the western region. We construct the regional dummy variable (western region = 1, eastern region = 0) and construct the interaction term between outward internationalization orientation, inward internationalization orientation, business ties, political ties and its square term and regional dummy variable. The analysis of results shows that compared with the eastern region, the influence of inward internationalization orientation of listed companies in the western region on firm performance is weaker ($\beta$ = -9.954191,t = −2.26,p<0.01); the effect of outward internationalization orientation on performance has no significant difference between the eastern and the western regions ($\beta$ = -.089045,t = -0.37, p>0.7). When the interaction between business ties and regional virtual variables is added to the regression equation, it is found to be insignificant ($\beta$ = -6.554928,t = -0.82,p>0.4), indicating that there is no significant difference in the mediating effect of business ties in the eastern and western regions. The results of the analysis of the U-shaped moderating effect of political ties found that the U-shaped moderating effect of political ties was more significant in the relationship between inward internationalization orientation and performance in the western region ($\beta$ = 907.9047,t = 3.95,p = 0.000), while there was no significant difference in the U-shaped moderating effect of political ties in the relationship between outward internationalization orientation and performance ($\beta$ = 1.325291,t = 0.65,p>0.5).

(3) Additional data. The data for the above analysis span the period of 2005–2013, in which the data for inward internationalization are manually compiled from the data on the equity structure of CSMAR listed companies. For data after 2013, the CSMAR database provides data on overseas direct investment, which includes overseas operating income, proportion of foreign investors and corresponding financial data of listed companies with overseas direct investment from 2013–2017. The data were combined with our manually processed data for 2005–2013 and added to 2017 for further robustness testing. The results are consistent with the hypothesis test. Among them, outward internationalization orientation has a positive effect on firm performance with significant results ($\beta$ = 0.223,$t$ = 2.76,$p$<0.01). After adding the role of business ties, the mediating variable business ties results are significant ($\beta$ = 4.819,$t$ = 2.63, $p$<0.01), and the role of outward internationalization orientation is still significant ($\beta$ = 0.211, $t$ = 2.61,$p$<0.01) but less than the effect of the first step. Business ties play a mediating role between outward internationalization orientation and firm performance. The results of the moderating effect of political ties show that political ties play a U-shaped moderating role in the relationship between outward international orientation and firm performance ($\beta$ = 6.954, $t$ = 2.65,$p$<0.01).

## 5.4 Discussion

We propose that in the relationship between internationalization and firm performance, social networks have a dual mechanism. The research results show that in the relationship between outward internationalization orientation and firm performance, business ties play a mediating role. In the inward internationalization orientation, the mediating effect of business ties was not found using the traditional three-step approach, but a partial mediating effect of business ties was found using the bootstrapping approach. We believe that the reason for this result may be that the impact of competition is ignored. In the inward internationalization

orientation, companies have not yet expanded overseas, and their internationalization approach is mainly through gaining the experience and technology of overseas companies locally, while business ties capture the situation in which companies gain international experience from other companies. In our study, the research perspective of business ties is based on local business relations. Therefore, there is a competitive and cooperative relationship between companies and their commercial partners. In recent studies, it has been found that competition and cooperation have an important impact on international joint ventures [53], but our study does not consider the role of competition and cooperation. Therefore, this may be the reason why the mediating role of business ties is not found in inward internationalization orientation. In general, the mediating mechanism of business ties is basically supported by data. From the theoretical level, the reason why business ties play a mediating role in the relationship between internationalization and firm performance is that business ties are conducive to corporate information acquisition and knowledge sharing [22], which is exactly the role played by the boundary spanners who assume the role of information processing [50]. Therefore, we use boundary spanning theory to explain why business ties play a mediating role.

Both outward internationalization orientation and inward internationalization orientation have been found to have a U-shaped moderating effect on political ties. The research results show that political ties have both positive and negative effects on companies. Due to the lack of an effective mechanism for long-term cooperation between companies and government officials [22], to overcome the negative effect of political ties in the process of corporate internationalization, companies need to establish strong political relations with government agencies to be able to promote corporate international performance. The comparative analysis of the eastern and western regions shows that while the mediating role of business ties in the eastern and western regions is not significant, the U-shaped moderating role of political ties in the eastern and western regions is different. In the western region, inward international-oriented companies are more affected by political ties. The possible reason for this outcome is that compared with the eastern region, the market system in the western region is less well developed, and the role of market competition is less pronounced than that in the eastern region. Therefore, the role of political ties in the western region is more obvious. In general, the moderating role of political ties is supported by data. From the theoretical perspective, political ties can provide both political legitimacy for the company [22], and guarantees in the face of external threats. If a firm has good government relations, it is conducive to promoting its internationalization strategy [48]. This is exactly the role played by the boundary spanners who assume the role of external representation [50]. Therefore, we use boundary spanning theory to explain why political ties play a moderating role.

In summary, we connect the business ties and political ties in social networks with information processing and external representation in boundary spanning theory and use boundary spanning theory to build a theoretical model of the dual mechanism of social networks. Based on boundary spanning theory, this paper explains the reasons for the dual mechanism of social networks.

# 6 General discussion

## 6.1 Conclusion

Through theoretical and empirical analyses, our research has led to the following conclusions. (1) In the relationship between internationalization and firm performance, there is a dual mechanism for the role of social networks, with differences in the mechanisms of action between business ties and political ties. (2) Business ties play the role of a mediating mechanism in the relationship between internationalization and firm performance. Business ties are

conducive to information acquisition and knowledge sharing and play the role of information processing in the relationship between internationalization and firm performance. (3) Political ties play a U-shaped moderating role in the relationship between internationalization and firm performance, which indicates that the political ties of the firm have a negative effect. If the political ties established by firms are not strong enough, then political ties may have the potential to limit firms' internationalization. When companies have strong political ties, this can facilitate the internationalization process.

The theoretical contribution of this paper comes from the following three aspects. (1) The proposal of the dual mechanism of social networks enriches the research on social networks, and promotes the existing research on the differentiation effect of social networks. Our results show that the differences of the different dimensions of social networks are not only in terms of the intensity of the role but also the different dimensions of social networks differ in terms of the mechanism of action. This result enriches the literature on social network research and provides a new perspective for future research on social networks. (2) Our study enriches boundary spanning theory by combining social network theory with boundary spanning theory. We apply social network theory's view that firm executives' individual-level social relations affect firm-level performance [19] and draw on boundary spanning theory's view that firm executives are boundary spanners [29] to combine social network theory with boundary spanning theory and apply boundary spanners' information processing and external representations to analyze the role of business ties and political ties in social networks. This research logic and the results of empirical research also contribute to boundary spanning theory. (3) Our research provides new evidence for research on multinational companies in emerging economies. Our empirical research based on Chinese listed companies supports the hypothesis of the dual mechanism of social networks, which provides new evidence for future research on the internationalization of companies in emerging economies.

The above research has the following implications in practice: First, the proposed dual mechanism model of social networks explains the role of business ties and political ties in the relationship between internationalization and corporate performance, and improves company managers' cognition of social networks, which prompts managers to recognize that there are differences in the mechanism of action between company business ties and political ties. This also helps managers make effective decisions on how to build corporate social networks. Second, according to the results of empirical analysis, the mediating role of business ties is significant in outward internationalization-oriented companies but not in inward internationalization-oriented companies, which suggests that managers should select business partners according to their company's internationalization strategy orientation. Finally, we find that political relations play a U-shaped moderating role in the relationship between internationalization and firm performance, which can help managers recognize that the role of political relations has two outcomes, both positive and negative. These results can provide a basis for managers to make decisions on how to build corporate political relations.

## 6.2 Limitations and future research directions

There are also deficiencies in this study. First, Chinese listed companies are selected as samples to test the model. Although the role of the industry is controlled for in the analysis process, it is not clear whether the role of social networks in different industries is different. Future research could analyze separate industries for more in-depth analysis. Second, we do not control for the impact of corporate ownership. Future research could examine the effect of corporate ownership on social networks in the process of internationalization. Finally, we combine social network theory and boundary spanning theory to construct a theoretical model of the dual

mechanism of social networks. Although the two roles of information processing and external representation based on boundary spanning theory explain the logic of the dual mechanism, the question of why a dual mechanism exists still needs further discussion at the theoretical level.

## Supporting information

**S1 Data.**
(ZIP)

## Author Contributions

**Conceptualization:** Xin Cao, Peng Li.

**Data curation:** Xin Cao, Limin Fan.

**Formal analysis:** Xin Cao, Peng Li, Limin Fan.

**Funding acquisition:** Xin Cao, Peng Li, Xiaozhi Huang.

**Investigation:** Xin Cao.

**Methodology:** Xin Cao, Peng Li, Xiaozhi Huang, Limin Fan.

**Writing – original draft:** Xin Cao.

**Writing – review & editing:** Peng Li, Xiaozhi Huang, Limin Fan.

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
