## [Decision Letter · Decision Letter 0]

16 Sep 2021

PONE-D-21-26838The Dual Mechanism of Social Networks on the Relationship between Internationalization and Firm Performance: Empirical Evidence from ChinaPLOS ONE

Dear Dr. Li,

Thank you for submitting your manuscript to PLOS ONE. After careful consideration, we feel that it has merit but does not fully meet PLOS ONE’s publication criteria as it currently stands. Therefore, we invite you to submit a revised version of the manuscript that addresses the points raised during the review process.

We look forward to receiving your revised manuscript.

Kind regards,

Elisa Ughetto

Academic Editor

PLOS ONE

Journal Requirements:

When submitting your revision, we need you to address these additional requirements. 1. Please ensure that your manuscript meets PLOS ONE's style requirements, including those for file naming. The PLOS ONE style templates can be found at https://journals.plos.org/plosone/s/file?id=wjVg/PLOSOne_formatting_sample_main_body.pdf and https://journals.plos.org/plosone/s/file?id=ba62/PLOSOne_formatting_sample_title_authors_affiliations.pdf
 2. PLOS requires an ORCID iD for the corresponding author in Editorial Manager on papers submitted after December 6th, 2016. Please ensure that you have an ORCID iD and that it is validated in Editorial Manager. To do this, go to ‘Update my Information’ (in the upper left-hand corner of the main menu), and click on the Fetch/Validate link next to the ORCID field. This will take you to the ORCID site and allow you to create a new iD or authenticate a pre-existing iD in Editorial Manager. Please see the following video for instructions on linking an ORCID iD to your Editorial Manager account: https://www.youtube.com/watch?v=_xcclfuvtxQ

Additional Editor Comments:

Dear authors, I agree with the reviewer that you have to add more details in yuor methodology, better discuss the findings and better link your findings with prior theory. Also please engage in a better discussion on the research gaps you want to adress.

Reviewers' comments:

Reviewer's Responses to Questions

**Comments to the Author**

1. Is the manuscript technically sound, and do the data support the conclusions?

Reviewer #1: Partly

2. Has the statistical analysis been performed appropriately and rigorously? 

Reviewer #1: Yes

3. Have the authors made all data underlying the findings in their manuscript fully available?

Reviewer #1: No

4. Is the manuscript presented in an intelligible fashion and written in standard English?

Reviewer #1: Yes

5. Review Comments to the Author

Reviewer #1: The research methodology should be improved

The analysis needs to indicate to details and implications should be based on these results

the introduction must indicate to main gaps and create a bridge between this part and literature review

6. PLOS authors have the option to publish the peer review history of their article (what does this mean?). If published, this will include your full peer review and any attached files.

Reviewer #1: **Yes: **Dariyoush Jamshidi

---

## [Author Response · Author response to Decision Letter 0]

27 Oct 2021

PONE-D-21-26838(R1): Response to Reviewers

Thank you very much for inviting me to submit this revision, and for your clear directions which I have used as main guidance. I appreciate your appraisal that my manuscript shows promise for making a valuable contribution. My point-by-point responses to your comments , are given below.

RESPONSES TO THE EDITOR’S COMMENTS

Additional Editor Comments: Dear authors, I agree with the reviewer that you have to add more details in yuor methodology, better discuss the findings and better link your findings with prior theory. Also please engage in a better discussion on the research gaps you want to adress.

Response: I have followed your suggestions, and do the following in the revised manuscript:

(1) I add more details in my methodology. I construct two econometric models, and I also introduce the methods of analyzing mediating effect and moderating effect.

(2) I further discuss the findings of the study. I emphasize that our research explains the reasons for the existence of dual mechanisms in social networks based on boundary spanning theory.

(3) In the introduction, I further discuss the relationship between current research and previous research. I explained that the existing research mainly focuses on the difference in intensity between business ties and political ties. However, the current research discusses the differences between the two from the perspective of impact mechanism. 

RESPONSES TO THE REVIEWER #1

Thank you for your careful review of my manuscript, and for your positive assessment that this paper has a potential to make a nice contribution to the literature. Thank you also for your excellent suggestions on how I can improve the paper. I show below how and where I have responded to your comments. 

The research methodology should be improved The analysis needs to indicate to details and implications should be based on these results the introduction must indicate to main gaps and create a bridge between this part and literature review.

Response: Thank you for these valid points. I have followed your suggestions, and do the following:

(1) In terms of research methods, I have constructed a mediating effect analysis model and a moderating effect analysis model to analyze the different roles of business ties and political ties.

(2) I introduced the details of the analysis method and explained the methods to test the mediating effect and moderating effect.

(3) I revised the introduction, briefly combed the research progress of the existing literature, and stressed that the existing research mainly focuses on the difference in strength between business ties and political ties. However, the current research analyzes the differences between the two from the perspective of impact mechanism, and explains the reasons for the differences based on boundary spanning theory.

(4) I further enrich the literature review of boundary spanning theory.

(5) I further discussed the results of the empirical study. In the discussion, I explained the support of the research results for the mediating effect of business ties and the moderating effect of political ties. Furthermore, I emphasize the role of boundary spanning theory in explaining the dual mechanism of social networks.

Revision information of reference list

In the revised manuscript, the references cited in the paper are also updated，the revised draft deleted two references and added 12 references. 

In the old manuscript, two references were deleted as follows：

12. Iurkov, V.; Benito, G.R.G. Change in domestic network centrality, uncertainty, and the foreign divestment decisions of firms. J Int Bus Stud 2020, 51, 788-812.

13. Cuypers, I.R.P.; Ertug, G.; Cantwell, J.; Zaheer, A.; Kilduff, M. Making connections: Social networks in international business. J Int Bus Stud 2020, 51, 714-736.

In the revised manuscript, 12 references were added as follows：

12. Horak, S.; Taube, M. Same but different? Similarities and fundamental differences of informal social networks in China (guanxi) and Korea (yongo). Asia Pac J Manag 2016, 33, 595-616.

14. Chen, C.C.; Chen, X.; Huang, S. Chinese guanxi: An integrative review and new directions for future research. Manage Organ Rev 2013, 9, 167-207.

15. Xiao, Z.; Tsui, A.S. When brokers may not work: The cultural contingency of social capital in Chinese high-tech firms. Admin Sci Quart 2007, 52, 1-31.

16. Wang, G.; Jiang, X.; Yuan, C.; Yi, Y. Managerial ties and firm performance in an emerging economy: Tests of the mediating and moderating effects. Asia Pac J Manag 2013, 30, 537-559.

17. Arnoldi, J.; Villadsen, A.R. Political ties of listed Chinese companies, performance effects, and moderating institutional factors. Manage Organ Rev 2015, 11, 217-236.

20. Zhang, M.; Qi, Y.; Wang, Z.; Zhao, X.; Pawar, K.S. Effects of business and political ties on product innovation performance: Evidence from China and India. Technovation 2019, 80, 30-39.

21. Du, J.; Zhou, C. Does guanxi matter in the foreign expansion of Chinese manufacturing firms? The mediator role of linking and leveraging. Asia Pac J Manag 2019, 36, 473-497.

23. Zhang, X.; Ma, X.; Wang, Y.; Li, X.; Huo, D. What drives the internationalization of Chinese SMEs? The joint effects of international entrepreneurship characteristics, network ties, and firm ownership. Int Bus Rev 2016, 25, 522-534.

24. Li, J.; Xia, J.; Zajac, E.J. On the duality of political and economic stakeholder influence on firm innovation performance: T heory and evidence from C hinese firms. Strategic Manage J 2018, 39, 193-216.

25. Shen, L.; Zhang, C.; Teng, W.; Du, N. How do business and political Networking shape overseas dispute resolution for state-owned enterprise from emerging economies. Int Bus Rev 2021, In Press.

36. Anderson, J.C.; Narus, J.A. A model of distributor firm and manufacturer firm working partnerships. J Marketing 1990, 54, 42-58.

37. Friedman, R.A.; Podolny, J. Differentiation of boundary spanning roles: Labor negotiations and implications for role conflict. Admin Sci Quart 1992, 28-47.

So in sum, I feel that the above point-by-point responses can answer and address your concerns. Thank you again for all your suggestions. They have helped me greatly in strengthening this paper. Thank you very much and best wishes!

---

## [Decision Letter · Decision Letter 1]

25 Jan 2022

The dual mechanism of social networks on the relationship between internationalization and firm performance: empirical evidence from china

PONE-D-21-26838R1

Dear Dr. Peng,

We’re pleased to inform you that your manuscript has been judged scientifically suitable for publication and will be formally accepted for publication once it meets all outstanding technical requirements.

Kind regards,

Elisa Ughetto

Academic Editor

PLOS ONE

Additional Editor Comments (optional):

Reviewers' comments:

Reviewer's Responses to Questions

**Comments to the Author**

1. If the authors have adequately addressed your comments raised in a previous round of review and you feel that this manuscript is now acceptable for publication, you may indicate that here to bypass the “Comments to the Author” section, enter your conflict of interest statement in the “Confidential to Editor” section, and submit your "Accept" recommendation.

Reviewer #1: All comments have been addressed

2. Is the manuscript technically sound, and do the data support the conclusions?

Reviewer #1: Yes

3. Has the statistical analysis been performed appropriately and rigorously? 

Reviewer #1: Yes

4. Have the authors made all data underlying the findings in their manuscript fully available?

Reviewer #1: Yes

5. Is the manuscript presented in an intelligible fashion and written in standard English?

Reviewer #1: No

6. Review Comments to the Author

Reviewer #1: The paper needs a professional proofreading because it will help to improve the paper for publishing.

7. PLOS authors have the option to publish the peer review history of their article (what does this mean?). If published, this will include your full peer review and any attached files.

Reviewer #1: **Yes: **Dariyoush Jamshidi

---

## [Editor Report · Acceptance letter]

2 Feb 2022

PONE-D-21-26838R1 

The dual mechanism of social networks on the relationship between internationalization and firm performance: empirical evidence from china 

Dear Dr. Li:

I'm pleased to inform you that your manuscript has been deemed suitable for publication in PLOS ONE. Congratulations! Your manuscript is now with our production department. 

Kind regards, 

on behalf of

Prof. Elisa Ughetto 

Academic Editor

PLOS ONE